Identification and functional analysis of bacteria in sclerotia of Cordyceps militaris

Luo Li 1 2
Zhou Jiaxi 2
Xu Zhongshun 2
Guan Jingqiang 2
Gao Yingming 2
Zou Xiao xzou@gzu.edu.cn 1 2
1 Institute of Fungus Resources, Guizhou University , Guiyang , Guizhou , China
2 Department of Ecology, College of Life Science, Guizhou University , Guiyang , Guizhou , China
Keller Nancy
Electronic publication date: 2021 Nov 25
Publication date: 2021
Volume: 9
Electronic Location ID: e12511
Received 2021 Aug 4; Accepted 2021 Oct 27
Copyright: ©2021 Luo et al.
Copyright year: 2021
Copyright holder: Luo et al.
License: This is an open access article distributed under the terms of the Creative Commons Attribution License, which permits unrestricted use, distribution, reproduction and adaptation in any medium and for any purpose provided that it is properly attributed. For attribution, the original author(s), title, publication source (PeerJ) and either DOI or URL of the article must be cited.
License URL: https://creativecommons.org/licenses/by/4.0/

Keywords: Cordyceps militaris, Microbial community, Co-culture, Functional analysis

Funding: The Science and Technology Department of Guizhou Province (Qian Ke He Zhi Cheng[2019] 2405) This study was funded by the Science and Technology Department of Guizhou Province (Qian Ke He Zhi Cheng[2019] 2405). The funders had no role in study design, data collection and analysis, decision to publish, or preparation of the manuscript.

==============================
Background

Cordyceps militaris is a fungus that parasitizes insects. Compounds from C. militaris are valuable in medicine and functional food. There are many kinds of bacteria in the natural sclerotia of C. militaris. However, the community structure of microorganisms in samples from different places may be different, and their corresponding ecological functions require experimental verification.

Methods

We used high-throughput sequencing technology to analyze bacterial 16S rRNA gene sequences in sclerotia of three samples of C. militaris from Liaoning Province, China. We isolated, identified and verified the function of culturable bacterial strains from the sclerotia.

Results

Pseudomonas, Pedobacter, Sphingobacterium, and Serratia were the dominant bacterial genera in the sclerotia. And function prediction showed that Pseudomonas and Pedobacter could be heterotrophic, Sphingobacterium could decompose urea, and Serratia could reduce nitrate. Two strains of bacteria isolated from the sclerotia of C. militaris, N-2 and N-26, were identified as Stenotrophomonas maltophilia and Pseudomonas baetica, respectively, based on culture and biochemical characteristics. When these isolated strains were co-cultured with C. militaris, the mycelium biomass and mycelium pellet diameter decreased, and the content of extracellular polysaccharide increased. Strain N-26 decreased the cordycepin content in C. militaris.

Conclusions

Bacteria in sclerotia have an important effect on the growth of C. militaris and the production of its metabolites.

Introduction

Cordyceps militaris is a member of the fungal genus Cordyceps and is used as a traditional Chinese medicine. It contains a variety of active substances including cordycepin (3′-deoxyadenosine), cordyceps polysaccharide, and cordyceps acid (Das et al., 2010). In humans, cordycepin and cordyceps polysaccharide improve immunity (Lee et al., 2020); protect the kidney (Han et al., 2020a); have antifatigue (Xu, 2016) and antioxidation properties (Song et al., 2015); inhibit bacterial growth (Ahn et al., 2000), inflammation (Zheng, Li & Cai, 2020), and tumors (Jin et al., 2018); and can be used as an effective anticancer supplement (Tuli et al., 2013). Cordycepin has been suggested for potential use against COVID-19 (Verma, 2020). Much attention has been paid to the cultivation and utilization of C. militaris (Jin et al., 2018; Lee et al., 2020).

C. militaris is a parasitic fungus that can infect larvae, pupae and adults of Lepidoptera, Coleoptera, Diptera and Hymenoptera (Shrestha et al., 2012; Xue et al., 2018); it is distributed throughout the Northern Hemisphere (Zhang et al., 2013). Because the host insects contain a variety of microorganisms, the sclerotia of Cordyceps also contain various microorganisms (Simon et al., 2019). In sclerotia of C. militaris collected in Yunnan Province, China, the bacteria identified included members of the phyla Proteobacteria, Acidobacteria, Bacteroidetes, and Actinobacteria, and the genera Pedobacter, Phyllobacterium, Pseudomonas, Mesorhizobium, Bradyrhizobium, Variovorax, Sphingomonas, and others (Zhang et al., 2021). The bacteria in sclerotia of C. sinensis were dominated by Proteobacteria and Actinobacteria and included Pseudomonas, Rhodoferax, Pedobacter, and Sphingomonas (Xia et al., 2016; Xia et al., 2019). In the sclerotia of Cordyceps cicadae, Proteobacteria, Bacteroidetes, and Actinobacteria were the main bacterial groups, and Pseudomonas and Serratia were dominant genera (Mou et al., 2021).

Some of these symbiotic/associated microorganisms have the ability to regulate the growth characteristics and metabolites of Cordyceps. Herbaspirillum and Phyllobacterium on the fruiting body can increase the bioactive compound content of C. militaris (Zhang et al., 2021). Three species of bacteria (Serratia marcescens, Cedecea neteri and Enterobacter aerogenes) isolated from C. cicadae promoted the production of N6-(2-hydroxyethyl) adenosine and decreased the production of adenosine, uridine and guanosine (Qu et al., 2019). In co-culture, the color of the fungus and the morphology of its mycelia may change (Tauber et al., 2016; Bor et al., 2016). The effect of microorganisms in the sclerotia on the growth of, and metabolite production by, C. militaris needs further research.

In nature, microorganisms coexist in complex communities that interact with each other (Hibbing et al., 2010). These interactions lead to the activation of otherwise silent biosynthetic pathways that affect the production of metabolites (Bertrand et al., 2014). Based on this principle, laboratories sometimes use co-culture to increase the accumulation of metabolites. The purpose of this study was to explore the interaction between the microbes in the sclerotia of wild and cultured C. militaris and the fungus. First, the microbial composition of wild C. militaris harvested in Liaoning Province, China, was analyzed; then, the bacteria were isolated from the sclerotia and identified. These isolated bacteria were then co-cultured with cultured C. militaris to study the effects of the bacteria on the morphology and biomass of mycelia pellets, and the yield of cordycepin and polysaccharide.

Materials and Methods

Sample preparation

Wild C. militaris was obtained in October 2019 from 12 insect pupa collected from soil in a broad-leaved mixed forest at an elevation of 240 m above sea level in Tieling City (42.39 N, 124.26 E), Liaoning Province, China. Cultured C. militaris (L.) Link was stored in the Institute of Fungi Resources of Guizhou University (GZUIFR; strain SYCM1910).

The wild C. militaris obtained from Liaoning Province was divided into three samples for analysis in this work. Sclerotia samples were prepared using the method reported in Zeng et al. (2019), with slight modifications. The sclerotia formed by C. militaris were rinsed with sterile water to remove residual soil, soaked alternatively with 75% alcohol and 2% sodium hypochlorite three times for 20 s each time, and then rinsed with sterile water. After removal from the body surface of the insect, the sclerotia were used for analysis. Each C. militaris sclerotia sample weighed about 3.5 g. The samples were stored at −80 °C until analysis.

Bacterial community determination by culture method

LB medium contained tryptone 10 g/L, yeast extract 5 g/L, NaCl 10 g/L, and agar 15 g/L (pH ≈ 7.0). Potato-dextrose-agar (PDA) contained potato 200 g/L, glucose 20 g/L, and agar 15 g/L (pH ≈ 7.0). Sabouraud’s medium contained glucose 40 g/L and peptone 10 g/L (pH ≈ 7.0).

Each C. militaris sclerotia sample (0.5 g) was ground, and its suspension was placed on Luria Bertani agar for microbial isolation at 25 °C. Then, bacteria isolated by the culture method were observed by scanning electron microscopy (SU8100, Hitachi), and their physiological and biochemical characteristics were identified using bacterial biochemical identification strips HBIG05 and HBIG08 (Qingdao Hopebio Biotechnology Co., Ltd.). Bacterial DNA was extracted according to the procedures for the Bacterial Genomic DNA Extraction Kit DP2002 (Beijing Bioteke Biotechnology Co., Ltd.). The 16S rRNA gene was amplified from all DNA extracts using primers 27F (5′-AGAGTTTGATCCTGGCTCAG-3′) and 1492R (5′-GGTTACCTTGTTACGACTT-3′) (Palkova et al., 2021). The reaction mixture (25 µL in total) contained 1 µmol/L primers (1 µL each), 10 ng/µL DNA template (2 µL), Master Mix (green) (including DNA polymerase, buffer, and dNTPs;12.5 µL, product number TSE005; Tsingke Biotechnology Co., Ltd.) and ddH2O (8.5 µL). The PCR conditions were: predenaturation at 95 °C for 3 min; 32 cycles of denaturation at 94 °C for 30 s, annealing at 55 °C for 30 s, and extension at 72 °C for 30 s; and a final extension at 72 °C for 10 min. PCR products were sequenced by Tsingke Biotechnology Co., Ltd. Using the sequence data, BLAST was performed against DNA sequences in GenBank, and the sequences of related species (similarity > 97%) were downloaded. Using Pseudomonas carboxydohydrogena as the outgroup, a phylogenetic tree was constructed by the neighbor-joining method using MEGA X software with 1,000 bootstrap replicates.

Co-culture of bacteria isolated from sclerotia

C. militaris SYCM1910 was inoculated on the center of a PDA plate and cultured at 25 °C for 7 days. Then, bacteria were inoculated at three locations on the periphery of the colony (25 mm from the point where C. militaris had been inoculated) and culture was continued at 25 °C for 7 days.

A piece of mycelium of C. militaris from a 21-day-old culture on PDA plate culture was inoculated into a 250-mL triangular flask containing 100 mL Sabouraud’s medium and cultured on a magnetic stirrer (120 rpm) at 25 °C for 3 days. Then, one mL/flask of bacterial suspension (bacterial cell density 1.5 × 108 colony-forming units/mL) was added and culture was continued at 120 rpm and 25 °C for 7 days. The co-cultured fermentation liquid was filtered using 0.45-µm and 0.22-µm microporous membranes, and then the filtrate was analyzed for the content of cordycepin and extracellular polysaccharides (EPS).

Dry weight of mycelium pellets was determined after the fermentation broth was filtered using qualitative filter paper and the pellets were dried to constant weight at 60 °C. Then, the mycelium pellet diameter was measured using vernier calipers.

Cordycepin content was determined by high-performance liquid chromatography according to the Agricultural Industry Standard NY/T 2116-2012 of the People’s Republic of China, using a Thermo Fisher Ultima 3000RS system and a C18 column with mobile phase acetonitrile: water (5:95 v:v) at flow rate 1.0 mL/min, column temperature 35 °C, detection wavelength 260 nm, and with sample volume 10 µL. EPS content was determined by the anthrone sulfuric acid method (Guo et al., 2016).

Bacterial community by non-cultural method

C. militaris sclerotia samples (3.0 g) were taken, ground in liquid nitrogen, and total microbial DNA was extracted according to the instructions of the E.Z.N.A.® SOIL DNA Kit (Omega, USA). PCR amplification used TransStart FastPFU DNA Polymerase. The reaction system contained: 5 × FastPFU buffer (4 µL), 2.5 mmol/L dNTPs (2 µL), 5 µmol/L primers 338F (5′-ACTCCTACGGGAGCAG-3′) and 806R (5′-GGACTACHVGGGTWTCTA-3′) (0.8 µL each) targeting the V3–V4 region of 16S rRNA genes (Zeng & An, 2021), FastPFU Polymerase (0.4 µL), bovine serum albumin (0.2 µL, 1 µg/µL), and template DNA (10 ng), supplemented with ddH2O to 20 µL. An ABI Gene AMP® 9700 PCR instrument was used for the reaction. The reaction parameters were 95 °C for 3 min; 30 cycles of 95 °C for 30 s, 55 °C for 30 s, and 72 °C for 45 s; and a final extension at 72 °C for 10 min. The amplified products were sent to Shanghai Major Biomedical Technology Co., Ltd. and sequenced using the Illumina MiSeq platform.

Paired-end reads obtained by MiSeq sequencing were first stitched by overlap, and sequence quality was controlled and filtered at the same time. Effective sequences were obtained by distinguishing samples according to barcode and primer sequences at both ends of the sequence, and sequence direction was corrected to obtain optimized sequences. Using UPARSE software (http://www.drive5.com/uparse/), repetitive sequence operational taxonomic unit (OTU) clustering was carried out with a threshold of 97% similarity, chimeras were removed in the process of clustering, and the RDP database (http://rdp.cme.msu.edu/) was used for OTU annotation.

The raw sequence reads obtained in this study were deposited in the NCBI Sequence Read Archive database under accession number PRJNA722375. FAPROTAX (http://www.zoology.ubc.ca/louca/FAPROTAX/), a tool that can predict ecological functions of bacterial and archaea taxa from 16S rRNA amplicon sequencing (Sansupa et al., 2021), was used to identify ecological functions of OTUs. An OTU abundance table and taxonomic annotation of OTUs were inputted in the corresponding option box, and the PLOT option was selected. Then, predicted function output was obtained as an Excel spreadsheet. A heatmap was generated using the online tool at the http://www.ehbio.com/ImageGP/.

Data analysis

Statistical analysis of the experimental data was performed using SPSS software v.22.0. The least significant difference test was used for one-way analysis of variance.

Results

Isolation and identification of bacteria

Two pure strains of bacteria, N-2 and N-26, were isolated from sclerotia of C. militaris. Strain N-2 is short rod-shaped (7.2–9.4  × 3.2–3.8 µm) (Fig. 1) and Gram-negative; colonies were slightly convex, pale yellow, smooth, moist, and opaque. Physiological and biochemical tests (Table S1) showed that strain N-2 is motile, can decompose glucose to produce pyruvate, and can decarboxylate the pyruvate and convert it into alcohol and other substances. In addition, ornithine decarboxylase, lysine decarboxylase and amino acid decarboxylase were detected, indicating that strain N-2 can decarboxylate amino acids (–COOH) to produce an amine and CO2. Strain N-2 cannot use mannitol, inositol, sorbitol, melibiose, ribitol, raffinose, xylose, or maltose as carbon sources. Using the methods described in the eighth edition of “Bergey’s Manual of Systematic Bacteriology”, strain N-2 was identified as belonging to the genus Stenotrophomonas. By BLAST analysis, the 16S rRNA gene sequence of strain N-2 was found to be 99.93% identical to that of S. maltophilia GZUIFR-YC01. Strain N-2 was identified as S. maltophilia (Hugh) (Fig. S1) (NCBI accession number: MW829549).

Figure 1 Scanning electron micrographs of strains.

(A) N-2 and (B) N-26. In (A), the reticular-like structure on the surface of the bacteria is secretion by the bacteria.

Strain N-26 is short rod-shaped (9.5–11.5 × 4.2–5 µm), Gram-negative, and its colonies are yellow, smooth, moist and opaque, with a central bulge. The semi-solid agar (dynamic test) of strain N-26 was positive, the Voges-Proskauer test was positive, and the Methyl Red test was negative. The strain was positive for ornithine decarboxylase, lysine decarboxylase and amino acid decarboxylase. The strain could not use mannitol, inositol, sorbitol, melibiose alcohol, raffinose, xylose, or maltose as carbon sources. In BLAST analysis, the 16S rRNA gene sequence of strain N-26 was 99.71% identical to that of Pseudomonas baetica YHNG5 (Fig. S1), which led to the identification of strain N-26 as P. baetica (Lopez) (NCBI accession number: MW829550).

Interaction between isolated bacteria and C. militaris

On PDA plates, S. maltophilia N-2 had an inhibitory effect on the growth of mycelium of C. militaris. Strain N-2 released something that slowed the growth of C. militaris mycelia near the area of S. maltophilia N-2 growth (Fig. 2A). P. baetica N-26 did not inhibit mycelial growth on PDA plates (Fig. 2B).

Figure 2 Co-culture on PDA plates.

(A) Strain N-2 and C. militaris, (B) strain N-26 and C. militaris, (C) C. militaris only.

The dry weight of mycelium pellets decreased from 1.57 g/flask in the control to 0.21 g/flask in the presence of S. maltophilia strain N-2 or 0.35 g/flask in the presence of P. baetica strain N-26 strain after co-culture with C. militaris for 7 days; these differences were highly significant when compared with the control (N-2: F = 885.476, P < 0.001; N-26: F = 493.275, P < 0.001) (Fig. 3A). The diameter of mycelium pellets decreased from 7.38 mm in the control to 3.29 and 3.63 mm after culture in the presence of strains N-2 and N-26, respectively (N-2: F = 1240.221, P < 0.01; N-26: F = 605.933, P < 0.001) (Fig. 3B). The addition of strain N-26 significantly decreased the cordycepin content of the culture medium from 3015.73 µg/g in the control to 2537.77 µg/g (F = 22.501, P = 0.009). However, addition of strain N-2 had little effect (3102.00 µg/g; F = 0.285, P = 0.622) (Fig. 3C). The EPS content was increased after bacteria were added to C. militaris culture, and the difference was significant compared with the control (N-2: 481.43 mg/g, F = 291.121, P < 0.001; N-26: 326.87 mg/g, F = 93.546, P = 0.001; control: 86.20 mg/g) (Fig. 3D).

Figure 3 Effects of coculture of isolated bacterial strains with C. militaris.

(A) Dry weight of mycelium pellets. (B) Diameter of mycelium pellets. (C) Cordycepin content in dry weight of mycelium pellets. (D) Polysaccharide content in dry weight of mycelium pellets. **P < 0.01.

Bacterial community composition and ecological function

A total of 204,067 valid sequences were detected in three sclerotia samples of C. militaris collected in October 2019 in Liaoning Province, China; 62,929–71,212 sequences were obtained for each sample, with average length 423.45–425.61 bp. With the increase of the number of sample sequences, the Shannon-Winner index curve flattened out (Fig. S2), indicating that the sequencing data depth in this experiment comprehensively reflected the bacterial community in the samples.

Bacteria identified in the sclerotia included 21 phyla, 46 classes, 123 orders, 195 families, 321 genera, 450 species, and 549 operational taxonomic units (OTUs). At the phylum level (Fig. 4), Proteobacteria (average relative abundance of OTU 68%), Bacteroidetes (24%), and Actinobacteria (8%) were dominant. At the genus level, Pseudomonas (17%), Unclassified Enterobacteriaceae (14%), Pedobacter (11%), Sphingobacterium (11%), Serratia (10%), Rhodococcus (6%), and Acromobacter (6%) were dominant.

Figure 4 Taxonomic composition of the microbiome in sclerotia of Cordyceps militaris.

Circles from inside to outside represent the community composition of the bacteria at different classification levels (kingdom, phylum, class, order, family, and genus, respectively). The size of the fan segments represents the relative proportion of the annotation results of different bacterial OTUs.

Forty-two OTUs were common to the three samples (Fig. S3), accounting for only 7.65% of the total number of OTUs. These 42 OTUs were uploaded to the FAPROTAX system for analysis, and predicted functions of 17 genera represented by 21 OTUs were identified (Fig. 5; Table S2). OTU1490 (Stenotrophomonas) is animal parasitic or symbiotic, and a human pathogen; it actively participates in the nitrogen cycle. OTU2342 (Sphingobacterium) is involved in urea decomposition. OTUs 1448, 2330 and 2314 (Pseudomonas) are chemoheterotrophic. OTUs 1539 and 1423 (Rhodococcus) degrade aromatic hydrocarbons and aliphatic non-paraffin hydrocarbons.

Figure 5 Functional prediction of the bacterial core microbiome in sclerotia of C. militaris using FAPROTAX.

Genera in red are the subject of this study.

Discussion

In our study, we isolated two bacteria belonging to the microbiota of C. militaris sclerotia, S. maltophilia N-2 and P. baetica N-26. Then, we co-cultured these bacteria with C. militaris, and found that both of them increased the EPS content of C. militaris, but Pseudomonas baetica N-26 decreased the cordycepin content.

Table 1 Relative abundance and function prediction of bacterial composition in sclerotia of C. militaris isolates.

Genus	C. militaris
from Yunnan (Zhang et al., 2021)	C. militaris
from Liaoning	Predicted function	
Pseudomonas	2.01%–15.00%	16.68%	aerobic chemoheterotrophy; chemoheterotrophy;	
Pedobacter	2.01%–15.00%	10.81%	aerobic chemoheterotrophy; chemoheterotrophy;	
Variovorax	2.01%–15.00%	3.14%	None	
Phyllobacterium	2.01%–15.00%	1.69%	None	
Labrys	2.01%–15.00%	0.98%	aerobic chemoheterotrophy; chemoheterotrophy;	
Mesorhizobium	2.01%–15.00%	0.05%	nitrogen fixation; aerobic chemoheterotrophy; ureolysis; chemoheterotrophy;	
Sphingomonas	2.01%–15.00%	0.06%	aerobic chemoheterotrophy; chemoheterotrophy;	
Bradyrhizobium	2.01%–15.00%	0.02%	None	
Serratia	——	9.65%	fermentation; nitrate reduction; chemoheterotrophy; plant pathogen;	
Achromobacter	——	6.26%	aerobic chemoheterotrophy; nitrate respiration; nitrate reduction; nitrogen respiration; chemoheterotrophy;	
Rhodococcus	——	5.78%	aromatic hydrocarbon degradation; aromatic compound degradation; aliphatic non methane hydrocarbon degradation; hydrocarbon degradation; chemoheterotrophy; ligninolysis; plant pathogen;	
Pantoea	——	4.06%	fermentation; mammal gut; animal parasites or symbionts; nitrate reduction; chemoheterotrophy;	
Luteibacter	——	3.89%	None	
Stenotrophomonas	——	1.82%	nitrate respiration; nitrate reduction; nitreogen respiration; aerobic chemoheterotrophy; human pathogens; animal parasites or symbionts ; chemoheterotrophy;	
Ochrobactrum	——	1.02%	None	
Notes.

“–” means not mentioned in the literature; “None” means that there was no result when a function was predicted using FAPROTAX.

The bacteria present in sclerotia of C. militaris sampled from Liaoning Province, China, were identified using high-throughput sequencing technology. Pseudomonas were more abundant in sclerotia of C. militaris isolated in Liaoning Province than in C. militaris isolated in Yunnan Province, while Phyllobacterium, Mesorhizobium, and Bradyrhizobium were less abundant in the former (Zhang et al., 2021). The relative abundance of Mesorhizobium, Bradyrhizobium, Sphingomonas, and Labrys in sclerotia samples from Liaoning Province was lower than in samples from Yunnan Province (Table 1). Pseudomonas, Pedobacter, Phyllobacterium, Mesorhizobium, Bradyrhizobium, Sphingomonas, Variovorax, and Labrys were found in C. militaris samples from both Yunnan Province (southwest China, 25.40 N, 102.92 E) and Liaoning Province (northeast China, 42.39 N, 124.26 E), but their relative abundances were different, which may be related to differences of the insect host and environmental conditions (Yun et al., 2014). The bacteria found in the sclerotia may be key microorganisms in the microenvironment of C. militaris and perform important functions (Lemanceau et al., 2017). Functional prediction (Table 1) showed that Pseudomonas, Pedobacter, Labrys, and Sphingomonas are chemoheterotrophic, while Stenotrophomonas functions in the nitrogen cycle. Pseudomonas, Phyllobacterium, Mesorhizobium, Bradyrhizobium, Pedobacter, Variovorax, and Sphingomonas belong to the microbiome of the plant rhizosphere (Etesami & Glick, 2020; Yin et al., 2020). These microorganisms may help to maintain plant hormone balance, control root development, promote nutrient acquisition, prevent disease, improve plant growth, and maintain plant health (Xu et al., 2018; Asaf et al., 2020). Such microorganisms are also ingested by root-feeding insects. When Cordyceps spp. invade insects, the microorganisms in the insect gut interact with the fungus (Lei et al., 2015; Zhong et al., 2014).

With regard to the potential ecological functions of the two strains we isolated, Stenotrophomonas maltophilia, a parasitic bacterium of insects (Gandotra et al., 2018), can promote the digestion and absorption of food by the host by secreting enzymes such as cellulase, amylase, protease, and chitinase. These enzymes inhibit the integrity of fungal hyphae and biofilm formation (Ali Huda, Hemeda & Abdelaliem, 2019; Jankiewicz & Brzezinska, 2015; Rossi et al., 2014). In insects, S. maltophilia can inhibit the growth of Beauveria bassiana (a fungus that parasitizes arthropods) (Zhou et al., 2018). Therefore, S. maltophilia can play a protective role in an insect host. S. maltophilia participates in the sulfur and nitrogen cycles, degrades complex compounds and pollutants, and promotes plant growth and health (An & Berg, 2018). We conclude that S. maltophilia plays an important ecological role in the sclerotia of C. militaris.

Pseudomonas has many functions, e.g., P. fluorescens secretes luciferin and inhibits the growth of Escherichia coli in insects (Roberts et al., 2018), decomposes wood, synthesizes multiple vitamins, and suppresses fungi in beetles (Peral-Aranega et al., 2020). P. aeruginosa strain BGF-2 isolated from German cockroach could inhibit the growth of B. bassiana (Huang et al., 2013). Pseudomonas has a flexible metabolism that allows it to synthesize a wide range of antibiotics to ward off competitors, protect itself from predators, and produce chemical signaling molecules that sustain intraspecies and interspecies interactions (Götze & Stallforth, 2019).

The relative abundance of bacteria in insects may be related to growth stimulation by parasitic fungi. For example, the biomass of P. fragi (a bacterium found in Thitarodes and Hepialus ghost moths) increased after invasion by C. sinensis, and P. fragi became the dominant bacterium and participated in the process of larval mummification (Wu et al., 2020). Both Stenotrophomonas and Pseudomonas have been reported to inhibit conidial germination and mycelial growth of B. bassiana (Zhou et al., 2018). The two bacterial strains tested in this study had a similar effect on the hyphae of C. militaris. These findings indicate that the use of parasitic bacteria to inhibit fungal invasion is a protective mechanism of insects.

In the present study, the biomass of C. militaris decreased after co-culture with strain N-2 (S. maltophilia) or N-26 (P. baetica), so the cordycepin content in the culture medium decreased accordingly. Several mechanisms could explain this effect. One is inhibition of the expression of cordycepin-synthesis-related genes (cns1, cns2, cns3, and cns4) (Zheng et al., 2011); this can be verified by quantitative PCR in later study. Alternatively, (a) key enzyme(s) involved in cordycepin synthesis may have been inhibited. It is also possible that the mycelium structure was damaged in the co-culture process, which could be observed by using cryo-electron microscopy after co-culture. Alternatively, cordycepin may be produced at around the usual level but used by the co-cultured bacterium.

Because cordycepin has antibacterial function (Jiang et al., 2019), we speculate that its ecological role is to inhibit the growth of some bacteria, to create good conditions for C. militaris and keep the insect host from decaying. Our data indicate that some symbiotic bacteria may inhibit the production of cordycepin. We speculate that other symbiotic bacteria may promote the production of cordycepin. Thus, we aim to isolate other microorganisms from the sclerotia and further explore the relationships between the isolates and C. militaris. When increasing numbers of interactions are revealed, the ecological functions of microbes in the sclerotia will become clearer.

The biosynthetic potential of many bacterial and fungal strains is much greater than previously thought. For example, in Pseudoalteromonas sp. MEBiC 03485 co-cultured with Porphyridium cruentum UTEX 161, the content of sulfated polysaccharide was increased (Han et al., 2020b). The mechanism may lie in the effect of some small-molecule elicitors on the related transcription of secondary metabolite gene clusters (Pettit, 2011). The results of a previous study suggest that the increased production of EPS in the present study may be related to the protein phosphoglucomutase (Wang et al., 2021). However, it is unclear whether the increase was in the original type of polysaccharide or in new types of polysaccharide; this requires further study.

Conclusion

The microbiota of the sclerotia of C. militaris contains a diversity of bacteria, among which Pseudomonas, Pedobacter, and Serratia are the dominant genera. This study reveals the interactions between C. militaris and isolated strains of S. maltophilia and P. baetica; these bacteria had inhibitory effects on the biomass and mycelial pellet diameter of C. militaris, and increased its EPS content. Furthermore, P. baetica strain N-26 decreased the cordycepin content in C. militaris. These results enrich the study of microbial interactions in entomogenous fungal microenvironments and provide reference for improving the use of metabolites.

Supplemental Information

Supplemental Information 1 Phylogenetic analysis of the bacterial strains isolated in this study

Note: The number at each branch point is the bootstrap percentage (1000 resamplings). Numbers in parentheses are GenBank accession codes. Bar: 2% sequence divergence.

Click here for additional data file.

Supplemental Information 2 Shannon rarefaction curves for bacterial communities

Click here for additional data file.

Supplemental Information 3 Venn diagram of OTU overlap in the three samples

Click here for additional data file.

Supplemental Information 4 Raw data of Fig. 3

Click here for additional data file.

Supplemental Information 5 Physiological and biochemical characteristics of strains N-2 and N-26

Click here for additional data file.

Supplemental Information 6 Classification, functional prediction and abundance of OTUs in sclerotia of C. militaris

Click here for additional data file.

Supplemental Information 7 Sequence of N-26

Click here for additional data file.

Supplemental Information 8 Sequence of N-2

Click here for additional data file.

We thank Chun-Bo Dong for some suggestion of the manuscript.

Abbreviations

C. militaris Cordyceps militaris

S. maltophilia Stenotrophomonas maltophilia

P. baetica Pseudomonas baetica

O. sinensis Ophiocordyceps sinensis

VP test Voges-Proskauer test

MR test Methyl rea test

BLAST Basic local alignment search tool

NCBI National center for biotechnology information

HPLC High performance liquid chromatography

HEA N6-(2-hydroxyethyl)-adenosine

SE Scan Electron microscopic

EPS Extracellular polysaccharide

ddH2O Double distilled water

LSD Least Significant Difference

ANOVA Analysis of Variance

Additional Information and Declarations

Competing Interests

Author Contributions

DNA Deposition

Data Availability

The authors declare there are no competing interests.

Li Luo performed the experiments, analyzed the data, authored or reviewed drafts of the paper, and approved the final draft.

Jiaxi Zhou and Jingqiang Guan analyzed the data, authored or reviewed drafts of the paper, and approved the final draft.

Zhongshun Xu analyzed the data, authored or reviewed drafts of the paper, provided these samples, and approved the final draft.

Yingming Gao analyzed the data, prepared figures and/or tables, and approved the final draft.

Xiao Zou conceived and designed the experiments, authored or reviewed drafts of the paper, and approved the final draft.

The following information was supplied regarding the deposition of DNA sequences:

The sequence of Strain N-2 is available at NCBI: MW829549 and the sequence of Strain N-26 is available at NCBI: MW829550.

The following information was supplied regarding data availability:

The raw data are available in the Supplemental Files and at NCBI.

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
