# Peer review of "Identification and functional analysis of bacteria in sclerotia of Cordyceps militaris"

_PeerJ, doi:10.7717/peerj.12511_

## Round 0.1 · original submission · Major Revisions

Three experts reviewed your work, their responses vary from major revision to rejection. I do think if you can address the concerns of these articles we can reconsider your work. However, this will take significant revision and a total overall of English usage.

Reviewer 1 ·

Basic reporting

1. The language of this manuscript needs to be polished.
2.The expression of research results needs to use the past tense. For example, "shows" in the “abstract” should be changed to "showed".
3. The sentences in lines 33-36 need to be language polished and relevant references need to be added.
4. The format of Table S1 needs to be modified. For example, please delete the blank rows in the Table S1.
5.The sentences on lines 237-239 need grammatical corrections.
6.On lines 265-266, "indicate" should be changed to "indicated", and "is" should be changed to "was".

Experimental design

1. The authors need to explain in detail what is the significance of this research.
2. In line 70 "The 12 C. militaris...", what does this "12" mean? Please explain in the manuscript.
3.Is the "102.92 W" on line 228 correct? I think it is wrong.

Validity of the findings

no comment

Additional comments

1. "Potato-dextrose-agar (PDA)" appears in line 202, but "potato-dextrose-agar" also appears in line 80. Therefore, "potato-dextrose-agar" in line 80 should be changed to "potato-dextrose-agar(PDA)", and "potato-dextrose-agar (PDA)" in line 202 should be changed to "PDA".
2. The “P” on line 209 needs to be italicized.

Reviewer 2 ·

Basic reporting

The English language and grammar in this manuscript should be greatly improved throughout. Please also see Detailed corrections and comments.

Line 15: “the corresponding ecological functions” should be “their corresponding ecological functions”.

Line 21: “shows” is changed to “showed”.

Line 22: “Serratia could reduce nitrate” should be “and Serratia could reduce nitrate”.

Lines 23-25: “Two strains of bacteria, N-2 and N-26, were isolated from the sclerotia of the C. militaris, cultured, biochemically characterized, and identified as Stenotrophomonas maltophilia and Pseudomonas” can be improved into “Two strains (N-2 and N-26) of bacteria isolated from the sclerotia of C. militaris were identified respectively as Stenotrophomonas maltophilia and Pseudomonas, based on the cultural and biochemical characters” .

Lines 37-38: “Much attention has been paid to the development of C. militaris (Jin et al., 2018; Lee et al., 2020).” should be “Much attention has been paid to the cultivation and ultilization of C. militaris (Jin et al., 2018; Lee et al., 2020).”.

Line 39: This statement is not accurate enough. In fact, Cordyceps militaris can parasitize insect larvae, pupae and adults.

Lines 39-41: These three sentences could be combined into one sentence.

Line 58: “The effect of microorganisms on the sclerotia of C. militaris needs further research.” could be replaced by “The effect of microorganisms in the sclerotia on the growth and metabolite production of C. militaris needs further research.”

Lines 59-63: The whole paragraph should be rewritten to make sense.

Line 65: “Sample source and treatment” should be “Sample preparation”.

Line 66: “Sample:” should be omitted.

Line 67: The latitude and longitude information should be followed “Tieling City”.

Line 69: (GZUIFR) (Strain SYCM1910).

Line 71: “In other words, each sample was a mixed sample.” Should be omitted.

Line 72: Sclerotia samples were prepared by the method of Zeng (Zeng et al., 2019) with slight modifications.

Line 73: “C. militaris was rinsed” or “The sclerotia formed by C. militaris were rinsed”? C. militaris is a fungus.

Line 75-76: “After removal of the body wall of the insect, the specimen was sclerotia.” is changed into “After removal of the body surface of the insect, the sclerotia was used for analysis”.

Line 78: “Culture media and isolation of bacterial strains” should be “Bacterial community by cultural method”. I would suggest that “Identification of bacteria isolated from sclerotia” section is combined into “Bacterial community by cultural method”, to make the description more clearly.

Line 84: “Total bacterial DNA extraction, PCR amplification, and high-throughput sequencing” into “Bacterial community by non-cultural method”.

Line 97: “Isolated bacteria were observed” into “Isolated bacteria by the cultural method were observed…”.

Line 115 and 120: The subtitles should be omitted.

Lines 116-118: The bacteria grew well on PDA? Usually PDA is not a medium for bacteria.

Line 121: A piece of mycelia of C. militaris from a 21-day PDA plate was inoculated…?

Line 122: on a magnetic stirrer at ? rpm? You should indicate the speed of the stirrer.

Line 125: “Measurement of physical indicators:” should be omitted.

Line 129: “Biochemical determination:” should be omitted.

Line 135: “data processing” section should be combined into “Bacterial community by non-cultural method” section in Line 84.

Line 147: “Prediction of bacterial function” section should be also combined into “Bacterial community by non-cultural method” section in Line 84.

Lines 148-150: “FAPROTAX (http://www.zoology.ubc.ca/louca/FAPROTAX/) is a tool that can predict ecological functions of bacterial and archaea taxa from 16S rRNA amplicon sequencing (Sansupa, et al., 2021). We used it to identify ecological functions of OTUs.” may be replaced by “FAPROTAX (http://www.zoology.ubc.ca/louca/FAPROTAX/), a tool that can predict ecological functions of bacterial and archaea taxa from 16S rRNA amplicon sequencing (Sansupa, et al., 2021), was used to identify ecological functions of OTUs.”

Line 154: “Data analysis and presentation” should be “Data analysis”.

Line 157: “Results and Analysis” should be “Results”.

Line 158: In the “Methods and materials” section, you descripted “Bacterial community by cultural method” first, so it is better that you introduced the results according to this order. So I would suggest that you put the “Isolation and identification of bacteria” section (in Line 179) first, then “Bacterial community composition and ecological function” section.

Line 159: “A total of 204,067 effective sequences were detected in three samples of C. militaris” is changed into “A total of 204,067 valid sequences were detected in three sclerotia samples of C. militaris”.

Line 190: “By Basic Local Alignment Search Tool (BLAST) analysis” should be “By BLAST analysis”.

Line 202: “On potato-dextrose-agar (PDA) plates” should be “On PDA plates”.

Line 209: F and P should be italic.

Lines 237-239: These microorganisms may help to maintain plant hormone balance, control root development, promote nutrient acquisition, prevent disease, improve plant growth and maintain plant health?

Lines 239-241: The microorganisms are also ingested by root-feeding insects. When Cordyceps spp. invade insects, the microorganisms in the insect gut interact with the fungus (references?).


In your supplemental files, why the dry weight of the three replicates in the group N-26 showed significant difference (Raw_data_of_Figure_5)? Please this problem with your data.

Line 292: the references should be well formatted. They are not acceptable in the present format.

Experimental design

Line 70: “The 12 C. militaris” meat what? 12 insect larvae or pupae infected by C. militaris? You should indicate the insect stage (larva or pupa). And which species of the insects were infected by C. militaris?

Line 82-83: You should indicate the temperature you used to culture the bacteria. For bacterial isolation, why you also used PDA medium which is usually used for fungal isolation?

Line 129: You should provide more details about the sample preparation of cordycepin analysis. You used the liquid medium co-cultured with C. militaris and a bacterial strain for cordycepin analysis by HPLC?

Line 180: Only two bacterial strains were isolated from the sclerotia of C. militaris by cultural method?

Lines 212-214: You stated that “The addition of bacteria decreased the cordycepin content of the culture medium from 44.04 μg/mL in the control to 6.55 μg/mL with strain N-2 or μg/mL with strain N-26 (N-2: F=342.421, P < 0.01; N-26: F=406.635, P < 0.001) (Fig. 5C)”. You should make sure that the cordycepin decrease in the culture medium was not caused by the decreased biomass of C. militaris mycelia. From the “Methods and Materials” section, I do not know clearly how you prepared the samples for HPLC analysis.

In the Results, the results on the physiological and biochemical description of two bacterial species from “Isolation and identification of bacteria” section are not so interesting, because you can identify the bacterial species only by molecular marker and the identified bacterial species are common in insect community.

Although the isolated N2 and N26 showed significant inhibitory effect on the growth of C. militaris on PDA plates and liquid Sabouraud’s medium, and the cordycepin content in the liquid medium decreased sharply, it is uncertain that the cordycepin decrease in the culture medium was not caused by the decreased biomass of C. militaris mycelia. I suggest you express the cordycepin content in the culture medium by μg per g dry weight of mycelium pellets, to evaluate the influence of the bacterial strains on the cordycepin content produced by C. militaris mycelia.

Validity of the findings

Lines 174-178: It seems to me that Stenotrophomonas (OTU1490), Sphingobacterium (OTU2342), Pseudomonas (OTU1448, OTU2330, and OTU2314) and Rhodococcus (OTU1539 and OTU1423) are big groups with many species in each genus, so the predicted functions unlikely cover different species in the genus. Please verify these statements.

Lines 279-280: You indicated that “Our data indicate that some symbiotic bacteria may inhibit the production of cordycepin.”. You have to express the cordycepin content in the culture medium by μg per g dry weight of mycelium pellets, to evaluate the influence of the bacterial strains on the cordycepin content produced by C. militaris mycelia.

Line 284: Inclusion should be rewritten according to the new analysis of the the cordycepin content in the culture medium by μg per g dry weight of mycelium pellets.

Reviewer 3 ·

Basic reporting

The authors of the manuscript are interested in Identification and functional analysis of bacteria in sclerotia of Cordyceps militaris. However, Zhang et al (2021) have compared the diversity and composition of the bacterial and fungal communities associated with naturally
occurring C. militaris, including sclerotia, stromata and the habitat soil and also the bacterial functions. The difference is the sample from different locations. In this Ms. the samples from Liaoning province have been used, and only bacteria in sclerotia was studied. There was little new results.
On the other hand, I think the research question is not meaningful. The infection of C. millitaris on insect is easily. The bacteria in sclerotia should be studied in other entomogenous fungus e.g.Beauveria bassiana and Metarhizium anisopliae which were been used in biocontrol.

Experimental design

no comment

Validity of the findings

no comment

---

## Round 0.2 · accepted · Accept

We are happy that you have successfully met the comments of the reviewer.

Reviewer 2 ·

Basic reporting

The manuscript has been greatly improved, especially in the language, method description and result experession.

Some minor errors:

1. Lines 159-160: Each C. militaris sclerotia sample weighed about 3.5 g. However, in Line 293, you indicated that “Each C. militaris sclerotia sample (0.5 g) was ground”. It seems that 3.5g is quite different from 0.5g.

2. Line 393: (Guo, et al., 2016) should be formatted.

Experimental design

No comments

Validity of the findings

No comments